# Community Resident Perceptions of and Experiences with Precarious Work at the Neighborhood Level: The Greater Lawndale Healthy Work Project

**DOI:** 10.3390/ijerph182111101

**Published:** 2021-10-22

**Authors:** Jeni Hebert-Beirne, Jennifer K. Felner, Teresa Berumen, Sylvia Gonzalez, Melissa Mosley Chrusfield, Preethi Pratap, Lorraine M. Conroy

**Affiliations:** 1Division of Community Health Sciences, School of Public Health, University of Illinois Chicago, Chicago, IL 60612, USA; 2School of Public Health, San Diego State University, San Diego, CA 92182, USA; Jfelner@sdsu.edu; 3Center for Health and Social Care Integration, Rush University System for Health, Chicago, IL 60612, USA; tesotran@gmail.com; 4Greater Lawndale Healthy Work Project, School of Public Health, University of Illinois Chicago, Chicago, IL 60612, USA; sgonza43@uic.edu; 5Lawndale Christian Health Center, Chicago, IL 60623, USA; mcmosley78@gmail.com; 6Division of Environmental and Occupational Health Sciences, School of Public Health, University of Illinois Chicago, Chicago, IL 60612, USA; plakshmi@uic.edu (P.P.); lconroy@uic.edu (L.M.C.)

**Keywords:** precarious work, qualitative research, community health, community-based participatory research, occupational health, social determinants of health

## Abstract

Work is a key social determinant of health. Community health and well-being may be impacted in neighborhoods with high proportions of people engaged in precarious work situations compounded by health inequities produced by other social determinants associated with their residential geography. However, little is known about how community residents experience work at the neighborhood level nor how work impacts health at the community-level, particularly in communities with a high proportion of residents engaged in precarious work. We sought to understand, through participatory research strategies, how work is experienced at the community level and to identify community interventions to establish a culture of healthy work. As part of a mixed-methods community health assessment, community researchers conducted focus groups with residents in two high social and economic hardship neighborhoods on Chicago’s southwest side. Community and academic researchers engaged in participatory data analysis and developed and implemented member-checking modules to engage residents in the data interpretation process. Twelve focus group discussions (77 community resident participants) were completed. Three major themes emerged: systematic marginalization from the pathways to healthy work situations; contextual and structural hostility to sustain healthy work; and violations in the rights, agency, and autonomy of resident workers. Findings were triangulated with findings from the concept-mapping research component of the project to inform the development of a community health survey focused on work characteristics and experiences. Listening to residents in communities with a high proportion of residents engaging in precarious work allows for the identification of nuanced community-informed intervention points to begin to build a culture of healthy work.

## 1. Introduction

There is no specific definition of precarious employment, but several investigators have described features or characteristics of precarious work, including unstable or temporary work, low wages, exposure to hazardous working conditions, insufficient or uncertain hours with irregular and unpredictable schedules, a high risk of unemployment, lack of access to social protection and economic benefits (e.g., a living wage or health and retirement benefits), little opportunity for advancement, legal and practical barriers to joining a union or bargaining collectively, limited protection from discrimination and exploitation, and powerlessness to exercise legally granted workplace rights [1,2,3]. Types of precarious employment include temporary work, direct hire on temporary labor contracts, hiring through temporary employment agencies, on call/daily hire work, contract work, outsourcing to other companies, independent contractors, and involuntary part-time work [4].

### 1.1. Health Impacts of Precarious Work 

An accurate estimate of non-standard employment, including precarious employment, is difficult, but several sources indicate that this type of work arrangement is increasing [4,5,6], especially in certain segments of the economy. African Americans and Latinx and those who are foreign-born are over-represented in precarious employment. Workers in precarious jobs have lower levels of education [5,7], experience higher levels of stress, and work in dangerous conditions that put them at greater risk of occupational injury and illness [7] and workplace fatality [7,8]. Precarious employment is associated with a deterioration in occupational health and safety (OHS) in terms of injury rates, disease risk, hazard exposures, and worker (and manager) knowledge of OHS and regulatory responsibilities [9].

Limited data suggest that precarious jobs are more dangerous and adversely impact health away from work. Precarious jobs are similar to unemployment in their adverse impacts on mental health, causing anxiety and depression, and may be more debilitating in the long run than having no job [10,11]. Research shows that the prevalence of anxiety and depressive disorders is similar among those who have jobs of poor psychosocial quality (i.e., jobs that have high demand, low control, are insecure, and have poor job esteem, defined as the level of respect and dignity an individual believes is associated with his or her job [12]) and those who are unemployed [13,14,15]. A review of international research found overwhelming evidence that, compared to full time work, precarious work is associated with worse psychological strain and higher physical injury rates [9]. One reason for these higher rates may be high levels of exposure to psychosocial risk factors and the association of these risk factors with occupational illness and injury [16,17,18,19,20,21,22]. Workers in precarious jobs are relegated to positions of low status and power; the associated tendency for low-status workers to be depersonalized, devalued, and taken advantage of [23,24] greatly increases exposure to psychosocial stressors that are detrimental to worker health, most notably harassment, bullying, and discrimination. These exposures may be more detrimental to the health of workers in precarious jobs compared to those in traditional workplaces because those workers are less likely to have access to traditional sources of protection (e.g., a formal employer, a human resources department, a supportive supervisor, a labor union), and thus are less likely to be able to end these harmful abuses. 

As others have observed, work as a determinant of health is complex but can be described as including both work arrangements and working conditions [1,25]. The nature of precarious employment leaves workers vulnerable to direct effects of the work arrangement (stress related to unemployment or employment insecurity) and indirect effects (adverse physical and psychosocial working conditions); it also makes them susceptible to the social factors outside of working conditions that lead to poorer health (insufficient or uncertain income, lack of access to health care and other social benefits) [26]. 

Addressing the health of people at work is now recognized as an important strategy to improve population health [27]; however, much attention is focused on interventions to reduce health hazards and promote healthy behaviors. Traditional approaches to workplace health and safety protections are employer- and worksite-based, as are traditional workplace health-promotion programs. Many workers with precarious jobs do not have a regular employer or worksite. For example, home care, domestic, and temporary workers may change jobs frequently, move in and out of employment, and work for multiple employers. 

The CDC/NIOSH Future of Work initiative prioritizes the impacts of precarious employment arrangements on the workplace, workers, and the workforce [28] The initiative promotes implementation of the Total Worker Health^®^ (TWH) approach that integrates protection from work-related safety and health hazards with the promotion of injury- and illness-prevention efforts to advance worker well-being while also underscoring the inextricable relationships that exist between these issues both on- and off-the-job, including those which involve workers’ families, communities, and society as a whole [29]. Current models of workplace culture of health [28,29,30,31,32,33,34,35] preclude communities of workers in non-traditional work settings, such as precarious work. Thus, while it is important to consider and advocate for workplace interventions, traditional workplace health and safety policies, programs, and practices may not be feasible for or reach the increasing number of people working in precarious jobs. 

Interventions to address precarious employment arrangements need to move beyond those directly involving the workers and even beyond workplaces or employers; they must address the upstream economic, political, and social drivers of precarious employment by intervening for institutional changes [27,36,37,38], and they must include the voices of workers from communities experiencing high rates of precarious employment. Therefore, efforts should be focused at the fulcrum of the local level. Neighborhoods and communities where precarious work is an important feature of the employment landscape are vital to understanding the application of TWH principles by synergizing worker health protection and worker health-promotion programs with community stakeholders: local organizations that advocate for worker protection, i.e., worker centers and similar community-based organizations and unions that represent workers with precarious jobs; employer agencies that hire people into precarious work; community- and faith-based organizations and local government agencies involved in community health promotion efforts; and local healthcare delivery organizations.

### 1.2. Precarious Work at the Neighborhood Level

Opportunities for work in major cities are significantly influenced by zip code due to systemic racism embedded in historical processes that segregated urban communities in the United States, like Chicago, by race, ethnicity, and income [39,40]. Compounded by power differentials related to social position (at the intersections of race/ethnicity, class, gender, and immigration status), residents in urban neighborhoods that are predominantly Black and Latinx have higher proportions of residents engaged in precarious work [41,42]. Community health and well-being may be impacted in neighborhoods with high proportions of people engaged in precarious work situations compounded by health inequities produced by other social determinants associated with their residential geography. Precarious work alone causes stress due to unpredictability of work and working hours, less time to take care of family members, and even less time for civic engagement [26,43]. Low-income workers are less likely to be able to refuse work that they know is unsafe and exploitative. Given how work opportunities can be structured by where you live and that increasing knowledge on how where one is born, lives, works, and plays impacts one’s health [44], neighborhoods are an underappreciated setting for worker health promotion. COVID-19 has had a significant impact on nearly all work situations but has most profoundly impacted those already in precarious work, resulting in sudden unemployment, severe underemployment, increase risk for exploitation, or sudden increased exposure to health hazards at work [45]. 

## 2. Background

### 2.1. The Greater Lawndale Healthy Work Project

The Greater Lawndale Healthy Work (GLHW) project in the University of Illinois at Chicago (UIC) Center for Healthy Work (a NIOSH Center of Excellence) is a community-driven, community-based participatory research (CBPR) project to determine the context of and barriers and pathways to healthy work in two high-hardship neighborhoods on Chicago’s southwest side, North Lawndale and South Lawndale (known to residents as Little Village), which together are referred to as “Greater Lawndale (GL)”. This participatory research project in its sixth year of development is an outgrowth of the Little Village Participatory Community Health Assessment (LVCHA). The LVCHA was established in 2010 to be a sustained, student-engaged, reciprocal community–academic partnership with organic community-based organizational leadership for community health inquiry. Issues relating to work and occupation as determinants of resident and neighborhood health consistently emerged across research components in the LVCHA, which led to the focus on promoting healthy work at the neighborhood level in the GLHW project. 

### 2.2. Partner Neighborhoods

North Lawndale and Little Village are two neighborhoods that are historically segregated. North Lawndale is comprised of 32,806 residents, with 86.8% identifying as Black. The median household income in North Lawndale is $28,897 compared to the Chicago median income of $55,703. About 14.41% of residents are unemployed [46]. Nearly half of North Lawndale and Little Village residents are not in the labor force compared to 33% in Chicago. A total of 55.7% of residents have no education beyond high school compared to 38.4% of Chicagoans [47]. Little Village has 72,091 residents and is the second most densely populated community area in Chicago. The median income is $33,622. Little Village ranks third highest in the percent of residents who identify as Latinx (82.6%), and nearly 40% are foreign born [46]. More than 76.5% of residents speak Spanish as their primary language at home [47]. About 9% of residents are unemployed, and 49.9% and 76% of residents have no education beyond high school [46]. 

North Lawndale is ranked 15th out of 77, and Little Village is ranked the 7th highest in social and economic hardship across the 77 community areas in Chicago [46] (Table 1). Both areas have relatively similar ratios between male and female populations. With respect to community social determinants of health indicators, GL experiences more crowded housing. The Chicago rate of crowded housing is 3.6%, whereas in LV, it is 10.4%. The life expectancy gap between North Lawndale and the rest of the city of Chicago is significant, as the Chicago average is 77 years, and the North Lawndale average is only 69 years old. Years of Potential Life Lost (YPLL) for the city is 8131, whereas North Lawndale has an estimate of almost double at 16,013 (LV at 4831) [46].

In terms of socio-economic indicators, individuals in Greater Lawndale (GL) report a higher proportion of employment in low-wage jobs that require less formal educational qualifications compared to the city of Chicago [48]. Residents of North Lawndale are mostly employed in retail trade and healthcare and social assistance, and residents of Little Village are mainly employed in manufacturing and accommodation and food services as well as administrative support and waste management. Some 80% of these jobs are in the private sector where more than 20% do not have any health insurance coverage. Non-standard work arrangements are associated with lack of benefits, low wages, and unstable employment with little to no safety regulation [48]. As such, they are largely unprotected by standard labor laws and workers’ and communities’ employment rights. Industries with the highest representation in GL also have higher incidences of injury and illness in comparison to average national and state (Illinois) injury/illness rate (Figure 1). When education level and individual earnings are considered, on average, individuals in the city of Chicago earn more money across all levels of education [47].

The opportunity for economic advancement is limited in GL. Lack of access to quality education, citizenship status, and language differences are barriers to obtaining healthy work. Additionally, low-wage jobs subject workers to the risk of occupational injury and illness, which can be devastating to workers and their families, particularly those already facing poverty. The lack of opportunity hinders an individual’s social position by influencing access to basic resources, such as food, housing, transportation, computing/internet, and other resources that may help them obtain healthy work. This continuous cycle of unhealthy work influencing opportunity and affecting healthy work has consequences for the general health and wellbeing of the neighborhood. 

This pattern is demonstrative of the structural violence that Chicago neighborhoods on the southwest side of Chicago, like Greater Lawndale, have experienced for decades. Violence is structural when it is in the form of invisible manifestations of violence, including harm that is built into the fabric of society, creating and maintaining inequalities within and between different social groups [49,50].

### 2.3. Greater Lawndale Healthy Work Academic–Community Partnership

This academic–community partnership [51], now almost a decade old, is comprised on the academic side of national leaders in occupational and community health sciences; an expert in psychosocial research on workplace harassment, discrimination, and worker health; and a legal scholar who brings a transformative justice framework, the solidarity economy—a set of economic activities that aim to prioritize social profitability instead of purely financial profits—for sustainably addressing the root causes of precarious work [52]. The team also includes post-doctoral fellows, staff, and students who are community residents and/or are bicultural and bilingual. On the community side, the partnership is currently led by the GLHW Council comprised of 17 community stakeholders bringing expertise in the community, community health work, faith-based leadership, worker centers advocacy, and community development. Community-Based Participatory Research approaches [53] offer grounded, mixed-methods, participatory, exploratory research strategies that involve researchers and community participants working together to identify the mechanisms through which precarious work impacts neighborhood health and change it for the better. 

In our research, we used a CBPR approach that strives to equitably involve community partners—community members and representatives of local community-based organizations and institutions—in such a way that all partners contribute their expertise and share the decision making and ownership of the research. CBPR allows for research reciprocity, leveraging existing strength and building capacity of both the community and the academic partner [53,54,55,56]. All of our activities are conducted in English and Spanish so that monolingual Spanish-speaking residents can participate. This includes simultaneous translation of our meetings and events by a professional translator who has been a GLHW partner since its origins. Community partners, residents, or representatives of GL serve as co-researchers, known as community researchers (CRs), at every stage in the research process. 

The GLHW project aims to build community capacity to recognize and promote worker health through the development and piloting of community-level interventions. The project uses a mixed-methods, exploratory, sequential study design involving interviews, focus groups, concept mapping, and community health-survey methodology to characterize the experience of residents with precarious work. To date, data have been used to map the employment landscape, build an evidence-informed Conceptual Model and a community-grounded Theory of Change, and develop community interventions via an action-mapping process resulting in a community-tailored Action Roadmap (manuscript in development). In this paper, we present findings from one of our mixed-methods research components—our focus group study with residents, which explored how residents perceive and experience work at the neighborhood level in a neighborhood with high social and economic hardship.

## 3. Materials and Methods

Within our mixed-method, exploratory, sequential study design, we identified the need for community-driven qualitative inquiry to explore how residents perceive and experience work at the neighborhood level in a neighborhood with high social and economic hardship. Our team chose focus group methodology for our exploratory research, as it is best suited to understand the rich quality of respondent interactions to exploring common/divergent trends and/or beliefs and exploration of social norms and cultural and institutional influences [57]. This study was reviewed by an ethics board and approved by the University of Illinois Chicago Institutional Review Board (IRB) assuring best practices in respect for persons, beneficence, and justice.

### 3.1. Instrument Development and Training 

Aligning with principles of CBPR, our academic–community partnership team co-developed the focus group guide emphasizing four domains of interest with respect to our research question: perceptions of health, barriers to work, challenges at work, and worker’s rights. We also asked about ideas for how to promote healthy work at the neighborhood level. We initially developed the focus group guide in English and then translated the instrument into Spanish. We tested the English and Spanish versions of the focus group guide in mock focus groups moderated by CRs serving as focus group moderators practicing each question and probe. When the questions on the Spanish focus group guide were not testing well (i.e., they were not understood well or failed to produce productive discourse), several of our Spanish-speaking CRs recreated the guide from scratch aligning FG questions with our overall research questions and domains. We then translated the final version of the guide from Spanish to English. 

We trained 10 CRs and 2 graduate students to conduct the focus groups using the finalized versions of the focus group guide. Trainings were conducted as in-person training in English and Spanish to ground the group in qualitative research and best practice in focus group methodology. Each focus group was moderated by two CRs and/or one CR and one graduate students. All moderators were training in human subjects research.

### 3.2. Recruitment

The CRs led the recruitment using an IRB-approved focus group script to identify groups of residents with a shared social context. Recruitment was focused on residents with knowledge of and experience with precarious work. Purposeful sampling approaches were employed. Participants were consented to participate in the focus group and were provided with a gift card for their time. All but two focus groups (n = 12 groups) took place in GL in neighborhood settings (i.e., a church, a community meeting space, a community-based organization).

### 3.3. Participatory Data Analysis

We used a constructivist grounded theory approach to analysis, which provided a flexible “set of principles and practices” for co-analyzing qualitative data (as opposed to stricter rules and requirements prescribed by traditional, non-participatory, grounded theory approaches [58]. In line with a constructivist grounded theory approach, we acknowledge that our understanding of the data is an interpretation, or construction, of the lived experiences and opinions of our participants and their communities rather than an exact representation. We believe this approach to analysis is most appropriate for answering our key research question given its emphasis on interpretation rather than representation as well as its focus on understanding processes and its utility in developing theories about phenomena of interest [58,59]. Rather than relying on a priori codes, as is common in other approaches to qualitative data analysis, in line with constructivist grounded theory, we used the process of memoing of audio segments to allow for the generation of a codebook. Coding is a systematic process of disaggregating text and reclassifying it by categories that represent views and experiences repeated within and across focus group interviews. Memoing entails making notations of researchers’ conceptual and theoretical insights relating to the themes [60].

We formed an academic–community qualitative data analysis think tank [61] to engage in participatory memoing as part of the data analysis process to develop the codebook. We determined that the audio versions of the focus groups were better suited to memoing than verbatim transcripts, as we would be able to keep the data in their original language and preserve emotion, emphasis, richness. This would also enable us to avoid losing meaning in the data that may be lost in transcribing and translation. Developing the codebook through our think tank allowed for group code identification and processing of emergent themes, leveraging the expertise of all members (community and academic), and attended to issues of trustworthiness of the analysis, i.e., credibility [62]. The final codebook contained a few deductive, a priori codes but consisted mostly of more interpretive, grounded codes. Based on our participatory, iterative analytic process, we determined that we had developed a codebook suitable to the data and that theoretical saturation had been met. We used Dedoose as our computer-assisted qualitative data analysis software. See Figure 2 for the steps to the analysis.

Multiple codes (including the first, second, and fourth author) were debriefed regularly to ensure intercoder reliability i.e., dependability. After data were coded, we selected analytic codes to examine. We looked at code patterns and relationships, including code frequency, code co-occurrence, and focus group type by code count. We drafted the three themes and cross-cutting theme [58,59,63]. Think tank members shared the emergent themes with broader academic–community research team, and adjustments were made to specify the role that relationships play and how experiences are embodied by residents.

Our community–academic team then developed member-checking modules (MCM) to test the trustworthiness, i.e., confirmability, of our findings by engaging residents in data interpretation. We also used the MCMs to identify best ways to “tell the community’s story.” Recognizing that the constructivist grounded theory approach to the data would result in rich, nuanced, descriptive insight that may not be amenable to distribution to community members, story typologies were created as a dissemination output to disseminate the findings, typically just disseminated in academic literature, to community residents (manuscript detailing the translation of research findings to story typologies pending). 

## 4. Results

### 4.1. Research Participants

Seventy-seven residents participated across 12 focus groups. Focus groups ranged from 5 to 12 participants, lasting between 38 min and 120 min. Participants included day laborers, street vendors, residents participating in English as a Second Language (ESL) and General Education (GED) programs, community health workers, church members, and members of a young adult group. Half had a high school degree or less, 36% were born outside the United States, and 41% spoke primarily Spanish (See Table 2).

### 4.2. Insights

Overall, focus group participants emphasized the importance of the need for healthy work to promote health in GL and described that residents’ negative experiences with seeking, getting, keeping, or dealing with issues at work impact their relationships, families, and the community as a whole. Three main themes emerged with one cross-cutting theme: Systematic marginalization from the pathways to decent work situationsContextual and structural hostility to sustain healthy workViolations in the rights, agency, and autonomy of resident workersCross-Cutting Theme: Strained social resources that exist as a consequence of inequities experienced by community residents with respect to work

*Text in italics* below are verbatim language from focus group participants themselves. See Figure 3 for a visual depiction of the analytic codes. The larger the font size, the more frequently the code was used. 

### 4.3. Systematic Marginalization from Pathways to Decent Work Situations

Focus group participants described obstacles to securing healthy work for community members that are structural and systemic. A participant explained:


*“There are problems larger than [Greater Lawndale] that are robbing residents of jobs.”*


Residents face discrimination in identifying jobs, needing to rely on word of mouth from family and friends, and also face discriminatory hiring practices by documentation status, language, criminal record history, age, gender, and race/ethnicity. A research participant reflected:


*“It’s racism…lets be real.”*


Relying on one’s social network for work is key (“*you have to know somebody to get in*”), but this leads to a common kind of work that is low wage, high risk, and unstable.

Lack of community resources to prepare residents for work were seen as the result of systems of oppression that result in economic and social disinvestment in Greater Lawndale. For example, participants noted the “*lack of investment in*” and poor quality of public education results in less skilled, prepared workers. There were some very good community resources noted, but they were described as insufficient, with few good job training opportunities in the community. It was also noted, especially among the focus group with young adults, that there are few healthy work mentors or apprentice opportunities, as most people are engaged in precarious work. 

An inherent community-level disadvantage exists with community contextual barriers to healthy work: lack of affordable, quality child care (familial resources are already strained); limited affordable, reliable transportation (workers are exploited by employer transportation options); concerns for safety in getting to and from work that restrict options for some work; job type availability (most work at the community level is precarious in nature); few good jobs existing in the neighborhood (disinvested employers, many who live outside of the neighborhood, “*go back to their mansions in the suburbs*”, and “*won’t*” hire residents). One participant explained:

*“There is just not the economic development* [in GL] *that is needed.”*

### 4.4. Contextual and Structural Hostility to Sustain Decent Work

Once secured, residents experience hostile work conditions that threaten job security. Contextual challenges are seen in the nature of work and conditions at work as well as the exploitive power of the employer. 

The nature of work described was precarious in the unpredictable hours, unknown terms, low wage, and long hours. Low wage and poor terms require more work and restrict consumption potential and family/leisure time. Work conditions were described as difficult, hard labor, and repetitive, with insufficient information and protective gear. Challenges due to the power of the employer (which can be used to exploit) include overt sexual, racial/ethnic, language, and documentation status discrimination as well as a culture that prioritizes profits over people, relying on a disposable workforce (e.g., “*someone else is waiting in line for this job*”) that does not need training or protection of rights. Some described no control at work in terms of safety, access to food and drink, and use of the bathroom as extreme exploitation that is not uncommon. Some community issues also threaten job stability. For example, fear and concerns for safety while traveling to jobs due to community crime and police harassment. 

### 4.5. Violations in the Rights, Agency, and Autonomy of Resident Workers

Focus group participants emphasized that workers overall do not know their rights. However, participants questioned the relevance of knowing one’s rights if (a) your human rights are already being violated, and (b) you are not able to exercise them without risking your job. Employers deliberately keep workers in the dark so as to keep them powerless. Employers face no consequences to their exploitation since it leads to profits. Dehumanizing types of work and work conditions or treatment at work were described as compromising one’s self-respect but endured to provide for themselves and their family. One focus group participant suggested that:


*“Nothing’s going to change…so I think it’s about just getting used to it even though it is not right.”*


Risk for exploitation prohibits self-advocacy. Healthy work as a human right brings dignity, satisfaction, and pride. When barriers are in place to healthy work, there are individual, family, and community health consequences. A research participant described that:


*“Dehumanizing work leads workers to think that they are paid pennies, so they must deserve pennies.”*


Power is central to experiencing work exploitation. The power of the employer to exploit is great, as is the power of the system to oppress and create barriers. One community member described the tension:

*“These are structural issues but with a very personal affect*—[we both] *struggle with* [this] *and feel powerless against [it].”*

### 4.6. Cross-Cutting Theme: Strained Social Resources as a Consequence of Inequities

Across the three themes, the issue of stress on the individual and their social networks emerged as a consequence of the inequities resident’ experience. The inability to get ahead despite excessive effort was exhausting and defeating. Impacts were described on one’s physical health both in terms of hard labor since there are few options but to take difficult work (“*aqui todos trabajamos*”) but also since one must ignore health due to lack of resources (health insurance) and time (cannot take off work or risk losing work) to care for oneself. This was depicted by one focus group participant explaining the phenomenon as follows:


*“Feels like you are running on a hamster wheel and you are sprinting and not getting anywhere.”*


Work-related stress results in family strain, with a worker either projecting frustration on one’s family or deliberately keeping stressors from their family to protect them, also described as “*bottling up*” feelings. Participants described the strain in this way: “*you can give every bit of yourself every single day and still not come out ahead*”. Excessive barriers to work and poor work conditions make informal, illicit work an option. 

Workers are influenced by community contextual issues to stay in precarious work situations due to devotion to family and social norms of unfair work. This creates chronic strain on the community. Participants described that the pathways to healthy work are too fraught, and rare opportunities for healthy work were insufficient and not the norm and are perceived as “*not for me*” and are avoided. Strong social bonds between residents engaged in exploitive work situations serve to build resilience, but the common referents in the neighborhood may also set lower expectations and reduce opportunities for group action against the exploitation. Social cohesion and capital are useful for getting and keeping jobs but may also hold residents back from speaking out against injustices. Disrupting systematic barriers to healthy work and creating good jobs for all while building resiliency and solidarity among residents were described as critical actions to promote health in Greater Lawndale.

## 5. Discussion

Work and, more specifically, access to healthy, stable work is central to the health and well-being of communities. Findings from our focus groups with residents of Greater Lawndale (n = 77) suggest that residents in this high-hardship area of Chicago experience unique exposures to and within precarious work situations as well as barriers to obtaining healthy work, thereby threating individual- and community-level health. These exposures and barriers are systemic, including hostile job situations that threaten job security, lack of awareness among residents of their workers’ rights, and violation of workers’ rights. While these experiences may be relatively unique to residents of Greater Lawndale relative to other community areas in Chicago, they align with exposures and barriers experienced by marginalized and minoritized people engaged in precarious work across work settings and jurisdictions at a national and international scale. For example, analyses of the nationally representative General Social Survey of adults in the United States find that precarity in work hours are ubiquitous across the labor market, but that workers who are Black, young, and without a college degree are the most exposed to precarious work, characterized by volatile work hours, limited schedule input, and short advance notice to work [64]. In addition, white workers who receive salaries are less likely than workers of other races/ethnicities, irrespective of salary/pay, to report little to no control over the number of hours they work. Both of these phenomena emerged as salient among participants across our focus groups.

Our data also demonstrate how the phenomena and context of precarious work strains community residents such that workers internalize or embody stressful work-related experiences, producing additional physical and mental health consequences. With so many people in the neighborhoods facing the same challenges, social norms are established that normalize, perpetuate, and sustain unjust and inequitable work experiences. Prior research consistently supports a link between precarious work and negative mental health at the individual level. [63,64]. Our study begins to elucidate this link at the neighborhood level. 

The drivers of precarious and lack of healthy work intersect at multiple levels of social ecology, including classism, sexism, ableism, and structural racism. Indeed, standard, healthy work was initially available only to men and white people [65]. Thus, interventions to address the problems discussed by residents of Greater Lawndale and advance worker equity and well-being require multi-level approaches that address intersecting systems of inequity (e.g., structural racism) as well as more proximal determinants of proximate determinants of precarious employment (e.g., lack of jobs in the neighborhood, access to reliable and safe transportation, workforce training programs) and health simultaneously [66]. 

It is important to note that these data were collected and initially analyzed prior to the emergence of the COVID-19 pandemic. COVID-19 has predictably served as a clarifying event illuminating occupational inequities and health inequities in these communities in which a large proportion of residents are at high risk for infection at work (due to their status as essential workers) and who are already experiencing or are at risk for economic instability. COVID-19 case and death rates have consistently been disproportionately high in the partner communities’ zip code (see Figure 2). Latinx and non-Latinx Black residents have been disproportionately affected by COVID-19 in Chicago, representing 40% and 33% of all COVID-19 Chicago deaths. From March 2020, there have been a total of 41,430 confirmed cases in Greater Lawndale zip codes. From March 2020, there had been a total of 777 deaths in Greater Lawndale zip codes [67] (Figure 4).

The experiences residents of Greater Lawndale described in the focus groups highlight many of the same experiences that precarious and essential workers reported during the pandemic. These include sudden unemployment, underemployment, and increased exploitation [43]. Furthermore, COVID-19 produced excessive stress and strain on both precarious and essential workers [67,68], something that residents of Greater Lawndale in our focus groups described as a regular occurrence.

Our findings complement increasing evidence that structural, upstream factors are responsible for the neighborhood level strains described [69,70]. Our study addresses a gap in research on how residents perceive and experience work opportunities in their neighborhoods. With increasing recognition of the need for qualitative research for health equity, few recent studies have examined perceptions of specific occupations within a neighborhood [71,72,73] or work transitions (due to age or gentrification) [74], but there remains little research on how work is experienced within a neighborhood. 

## 6. Limitations

While our participatory approach to analyzing the audio data allowed for more equitable engagement of partners in the analytic process, academic researchers identified the audio segments to be memoed by the think tank members. This may have resulted in a biased collection of audio segments. However, of note were two conditions that addressed this potential limitation: several of our academic team members are also community members, and focus group moderators were community members and were able to request certain segments of the focus group to be memoed. Academic–community partnership team members who were monolingual English speakers were unable to participate in the data analysis that happened within Spanish-speaking groups. Focus groups were intended to be 90 min long. However, the average focus group was shorter, lasting only 72 min. Another limitation was the reliance on convenience sampling of residents. Given the influence of shared social context, it is possible that different stories would be heard if a randomized approach to sampling and recruitment had been employed.

### Iterative Research Design

The findings from the focus group study examining how residents perceive and experience work at the neighborhood level were triangulated with findings from our concept-mapping study examining residents’ perceptions on how work impacts health in GL. The combined findings informed the sections of the 119-item community health survey that provided more detailed description of work than is amenable through qualitative methods (manuscript under review). 

The experience of the CR in serving as focus group moderators also led to the adaption of a trauma-informed approach to our iterative research, as the discussion of experiences with work in GL that were perceived as unfair and unjust became emotional for several groups (manuscript in preparation). 

The findings from the focus groups triangulated across the other research methods informed the intervention-mapping process that resulted in an Action Roadmap for community intervention (manuscript in development). We have launched three of the six primary community interventions identified: an evidence-informed, historically grounded, worker-justice-centered *loteria* game (a traditional Mexican card game, similar to Bingo) to integrate into a community awareness campaign to shift perceptions and beliefs on how work impacts health, especially for workers in precarious employment situations; a local effort to influence accountability of temporary staffing companies; and conducting and evaluating an educational workshop for community members on opportunities to create wealth through worker cooperatives. Additionally, we launched a storytelling project in collaboration with youth residents to better understand how COVID-19 has impacted the lives of workers in GL. 

## 7. Conclusions

We found that residents describe a sense that they are systematically excluded from healthy jobs and that existing jobs in the neighborhood that can be hostile job situations that threaten job security. We also found that residents are overall unaware of their workers’ rights and/or unable or feel powerless to exercise their rights without risking job loss. Social norms that normalize unjust work experiences put an emphasis on individual change. Normalization of work requires systems-level change and culture shifts at upstream levels to effect change at individual levels. Our community-driven research agenda is positioned to continue to explore the pathways through which work situations impact neighborhood health. Community-driven interventions that promote healthy work at the neighborhood level by building power, capacity, and equity can result in the systems change needed such that everyone has access to fair and healthy work. 

## Figures and Tables

**Figure 1 ijerph-18-11101-f001:**
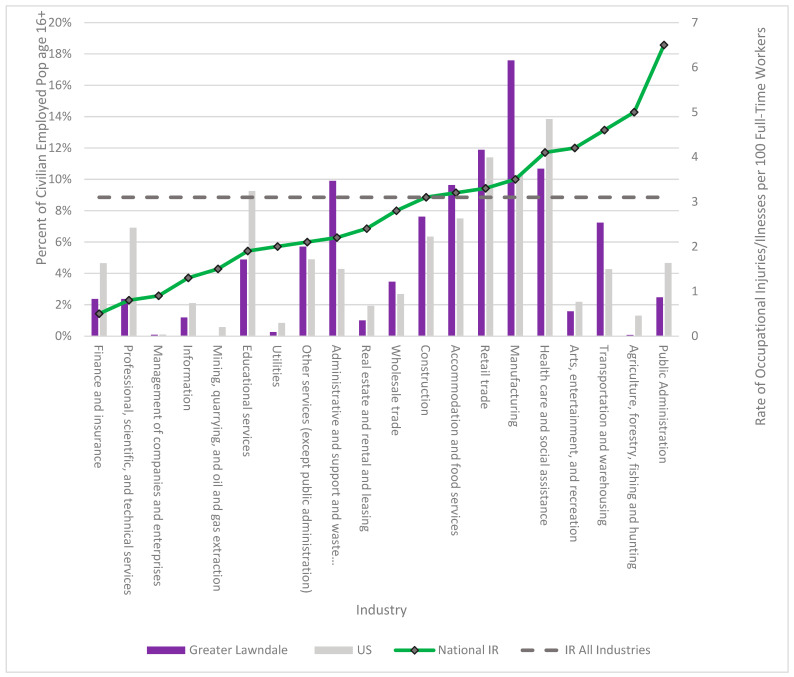
Industry Employment for Greater Lawndale and the United States compared to the Occupational Injury and Illness Rate for the United States.

**Figure 2 ijerph-18-11101-f002:**
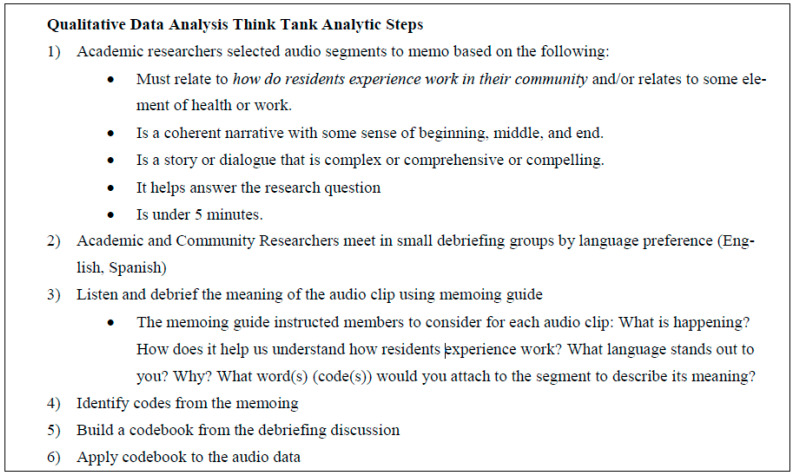
GLHW Qualitative Data Analysis Think Tank Analytic Steps.

**Figure 3 ijerph-18-11101-f003:**
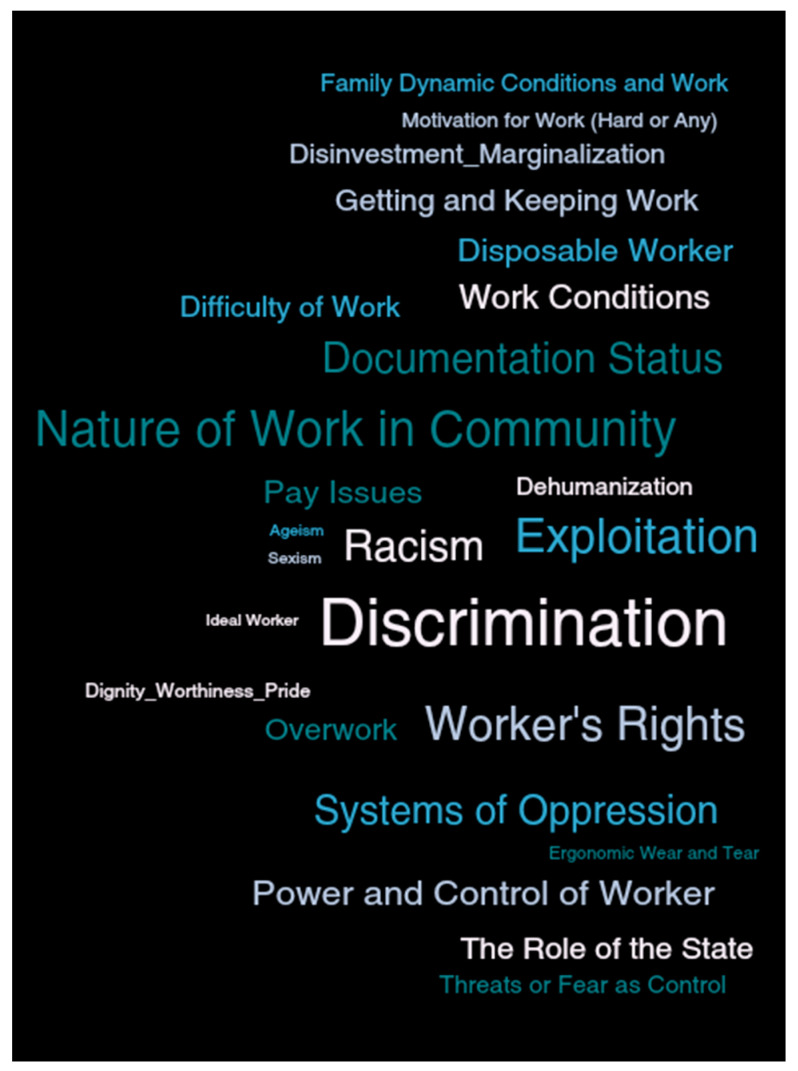
Analytic Code Word Cloud.

**Figure 4 ijerph-18-11101-f004:**
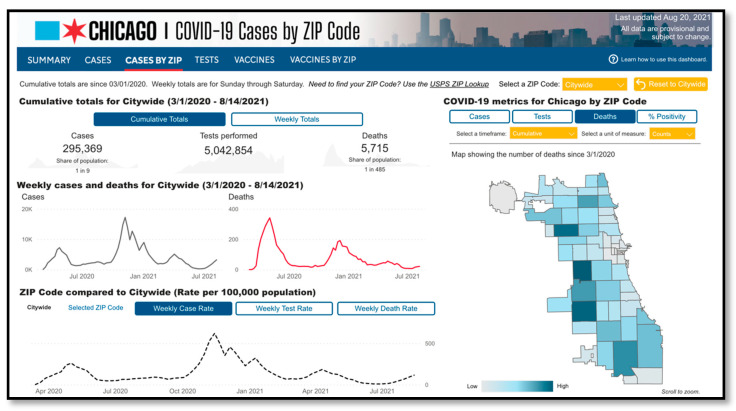
COVID Cases and Deaths in Greater Lawndale (zip codes 60623, 60624, 60608, 60612, 60632).

**Table 1 ijerph-18-11101-t001:** Comparison of Socio-demographic Characteristics of North Lawndale, Little Village, and Chicago.

Category	Indicator	Units	Time Period	North Lawndale	North Lawndale Margin of Error	Little Village	Little Village Margin of Error	Chicago	Chicago Margin of Error
Demography	Foreign born	% of residents	2015–2019	4.08	1.18	39.92	2.79	20.64	0.11
Demography	Limited English proficiency	% of residents	2015–2019	1.56	1.23	21.2	2.59	7.64	0.08
Population	Non-Hispanic White	% of residents	2015–2019	4.52	1.72	3.98	0.85	33.28	0.1
Population	Non-Hispanic Black	% of residents	2015–2019	86.8	5.9	12.9	2	30.1	0.2
Population	Latinx	% of residents	2015–2019	9.4	0.2	82.6	4.1	29.3	0.2
Population	Asian or Pacific Islander	% of residents	2015–2019	0.2	0.2	0.1	0.2	6.7	0.1
Population	Population	residents	2015–2019	32,086	1964	72,091	3254	2,709,534	66
Quality of Life	Years of Potential Life Lost (YPLL)		2013–2017	16,013	159	4831	58	8131	29
Quality of Life	Life expectancy	years	2019	69.1		78.9		77.3	
Infectious Disease	COVID-19 deaths	deaths	2020	77		194		4704	
Infectious Disease	COVID-19 death rate	rate per 100,000 population	2020	240.1		268.9		174.6	
Housing & Transit	Crowded housing	% of occupied housing units	2015–2019	3.19	1.94	10.41	4.67	3.62	0.09
Housing & Transit	Rent-burdened	% of renter-occupied housing units	2015–2019	58.63	6.17	60.31	13.17	45.97	0.32
Housing & Transit	Vacant	% of housing units	2015–2019	24.51	2.71	12.89	3.73	12.16	0.14
Education	High school graduation rate	% of residents	2015–2019	77.56	5.59	56.33	4.16	85.12	0.26
Education	College graduation rate	% of residents	2015–2019	12.5	1.91	7.85	2	39.48	0.18
Education	Preschool enrollment (0–4 years)	% of toddlers	2015–2019	53.35	26.04	52.21	13.24	57.89	0.92
Employment	Unemployment rate	percentage	2015–2019	14.41	3.64	8.7	5.32	8.06	0.1
Income	Hardship Index	score	2015–2019	88.7		94.1		62.2	
Income	Median household income	Income	2015–2019	$28,897	$4275	$33,622	$5185	$55,703	$342
Income	Per capita income	Income	2015–2019	$14,849	$1261	$12,364	$1109	$35,482	$195
Income	Poverty rate	% of residents	2015–2019	38.14	5.17	27.97	6.29	18.39	0.19
Public Assistance	Food stamps (SNAP)	% of households	2015–2019	44.6	4.19	24.36	5.58	18.26	0.15
Public Assistance	Households in poverty not receiving food stamps	% of households below the poverty line	2015–2019	34.69	6.98	47.43	13.49	48.05	0.42

**Table 2 ijerph-18-11101-t002:** Focus Group Participant Characteristics.

	Total		Little Village		North Lawndale		Other	
	N	%	N	%	N	%	N	%
Community Area	77	100%	40	52%	34	42%	3	6%
Country of birth								
U.S.	40	52%	5	13%	29	91%	4	80%
Mexico	35	45%	32	80%	2	6%	1	20%
Other	1	1%	0	0%	1	3%	0	0%
Missing	1	1%	1	3%	0	0%	0	0%
Language spoken at home								
Spanish	28	36%	27	68%	2	6%	0	0%
More Spanish than English	4	5%	6	15%	1	3%	1	20%
Both equally	8	10%	5	13%	3	9%	0	0%
More English than Spanish	8	10%	1	3%	3	9%	0	0%
Only English	29	38%	1	3%	23	72%	4	80%
Education								
Less than HS	19	25%	17	43%	2	6%	0	0%
HS graduate	19	25%	13	33%	6	18%	0	0%
Some college	8	10%	4	10%	4	12%	0	0%
College graduate or higher	31	40%	6	15%	22	65%	3	100%

## Data Availability

The data are not publicly available due to the fact that they are audio files and voice is a identifier. Further, research participants did not consent to having their voice shared publicly.

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
