# Peer review of "Community Resident Perceptions of and Experiences with Precarious Work at the Neighborhood Level: The Greater Lawndale Healthy Work Project"

_ijerph, 2021, doi:10.3390/ijerph182111101_

Round 1
Reviewer 1 Report
Some comments are suggested:
- The abstract must be structured with the sections of the manuscript specifying introduction, clearly summarized background, methodology, results, discussion and main conclusion.
- It would be interesting if the keywords are DeCS / MeSH descriptors.
- It is advisable to reduce the abstract as it is very long and not to use abbreviations.
- The introduction is interesting. Despite this, I consider that it has been extended a lot in explaining the context but little has been described about the published evidence on similar studies in different places, which may help to better understand the phenomenon under study and the research proposal. In addition, aspects that belong to methodology rather than introduction are observed (for example: lines 187-196). Also, the last paragraph of the introduction should be the overall objective of the study.
- In the design, reference is made to the fact that a mixed study has been developed. I think this is incorrect after full reading. Only one qualitative study was carried out, although sociodemographic data were extracted from the participants. It should be clarified.
- The methodology should include a subsection of ethical aspects. In it, reference must be made, among other things, to the authorization of an ethics committee or what was done with the recordings once they were transcribed.
- Was a program used for categorization and coding such as NVIVO or Atlas.Ti?
- What strategies of methodological rigor were carried out besides the triangulation like a transferability, credibility ...
- Was it taken into account for the selection of the groups, that the theoretical saturation of the data was reached? Clear out.
- The results must be improved. It is very confusing not to distinguish clearly between the definition of the codes and the verbatims even though they are in italics. It is advisable to separate them clearly from the text. Furthermore, the verbatims used are too concise and do not clearly represent the definition of the categories.
- The grounded theory has been used as a qualiative design. Despite this, the discussion has not established any connection with a theoretical orientation that can help the construction of theory from the analysis of the data. The results seem more like a simple description and it would take a higher level of abstraction to carry the theory. As stated, it would be an ethnographic or phenomenological study.
- The conclusions section should not include bibliographic citations, as these are the conclusions reached by the authors based on their data and comparison with similar studies.
- The literature used is somewhat obese and old.
Author Response
Dear Reviewer #1,
Thank you for your thorough and helpful feedback on our manuscript. We particularly valued the push to center our research in the latest evidence for how precarious work impacts health. See below our response to each of your concerns.
Issue |
Response |
Page/line |
1.1 The abstract must be structured with the sections of the manuscript specifying introduction, clearly summarized background, methodology, results, discussion and main conclusion.
|
The abstract has been structured per Reviewer 1’s suggestion. We’ve followed the IJERPH instuctions for authors that the abstract should be a single paragraph and should follow the style of structured abstracts, but without headings. |
Page 1/ lines 22-40 |
1.2 It would be interesting if the keywords are DeCS / MeSH descriptors.
|
Thank you for this advice. Now all terms are DeCS or MeSH except for precarious work and community health which we feel were important to maintain. |
Page 1/ line 42 |
1.3 It is advisable to reduce the abstract as it is very long and not to use abbreviations.
|
Abbreviations have been removed and we have shortened the abstract. |
Page 1/ lines 22-40 |
1.4 The introduction is interesting. Despite this, I consider that it has been extended a lot in explaining the context but little has been described about the published evidence on similar studies in different places, which may help to better understand the phenomenon under study and the research proposal. In addition, aspects that belong to methodology rather than introduction are observed (for example: lines 187-196). Also, the last paragraph of the introduction should be the overall objective of the study. |
This advice was helpful. We expanded the Introduction and added a separate Background section. We made sure the research objective was last paragraph in the Introduction section. We didn’t move the suggested lines since we felt like they presented the background needed (our existing community-based participatory research infrastructure [ CBPR]) prior to the methods, since this is our research approach. We also moved a section on COVID context to the discussion following this reviewer’s concern. This required moving Figure 2 to Figure 4 and changing Figure enumeration. |
Pages 1-2 Lines 45-108
Page 7 Lines 231-234
Page 14/ Lines 511-520 |
1.5 In the design, reference is made to the fact that a mixed study has been developed. I think this is incorrect after full reading. Only one qualitative study was carried out, although sociodemographic data were extracted from the participants. It should be clarified.
|
Thanks for letting us clarify. This manuscript describes the findings of one qualitative component of our iterative, mixed methods research study. |
See Page 7 line 232 |
1.6 The methodology should include a subsection of ethical aspects. In it, reference must be made, among other things, to the authorization of an ethics committee or what was done with the recordings once they were transcribed.
|
We’ve drawn attention to the IRB statement on line 209 and our adherence to research ethics. Note: we analyzed the audio files using Dedoose, qualitative data analysis software rather than transcribing which would result in losing emotions and emphasis |
Page 8 line 242-245 |
1.7 Was a program used for categorization and coding such as NVIVO or Atlas.Ti?
|
Thank you for asking for clarification. We have noted on line 263 that we used Dedoose. |
Page 9 Line 300 |
1.8 What strategies of methodological rigor were carried out besides the triangulation like a transferability, credibility ...
|
In addition to checks for trustworthiness in the form of member checking that we did with GLHW Council members and community members (ensuring confirmability), we described in Figure 2 the academic-community participatory think tank that co-developed codes from memoing of audio clips (ensuring credibility). We added description on to explain our attention to interrater reliability (ensuring dependability). In addition, we implicitly address transferability in the Discussion section of the manuscript by situating our findings in the extant literature and drawing conclusions about how our findings support, or diverge from, other research and how our study’s findings can advance worker health equity in similar communities and contexts with high rates of marginally employed or precarious workers |
Page 9-10 289-320 |
1.9 Was it taken into account for the selection of the groups, that the theoretical saturation of the data was reached? Clear out.
|
We reference theoretical saturation on line 288. |
Page 9 Line 288 |
1.10 The results must be improved. It is very confusing not to distinguish clearly between the definition of the codes and the verbatims even though they are in italics. It is advisable to separate them clearly from the text. Furthermore, the verbatims used are too concise and do not clearly represent the definition of the categories. |
We tried to utilize best practice in communicating qualitative findings, as noted by Reviewer 2. We’ve provided edits to better distinguish verbatim textual evidence of our thematic findings, to address concerns of Reviewer 1 |
Page 10-14 Lines 322-472 |
1.11 The conclusions section should not include bibliographic citations, as these are the conclusions reached by the authors based on their data and comparison with similar studies. |
Citations have been removed from the conclusion section and moved to the discussion section. |
|
1.12 The literature used is somewhat obese and old. |
We have updated the literature in the introduction section. |
|
Thank you again for your review.

Reviewer 2 Report
The article lacks a broad systematic review of the literature. The theoretical part is poorly described. This is also indicated by the small number of references cited in this part.
The research part is correct. Interesting topic and interesting results
Introduction: - there is no systematic review of the literature, i.e. defining individual concepts - so far conducted research on this topic, or the lack of it, which can also be shown by analyzing such databases as PRPQUEST or SCOPUS, etc. - some paragraphs do not have sources listed - the text of the introduction is actually not a scientific introduction, but rather a description of the research area.
Materials and Methods - here everything is correct - hence my conclusion that the authors are practitioners rather than scientists
Results Clear and comprehensible description of the results, also very well presented graphically.
Conclusion Good summaries. You can add information about the possibility of using the research results or plans for the future continuation of the research topic.
In conclusion, the topic of the articles is important and innovative. the authors lacked the ability to work on the achievements of other researchers or did not take it into account in the text.
After making changes, the article can be published.
Author Response
Thank you for your thoughtful review of our manuscript, Community Resident Perceptions of, and Experiences with, Precarious Work at the Neighborhood Level: The Greater Lawndale Healthy Work Project to the special issue Worker Safety, Health, and Well-Being in the USA. Below we address each reviewers’ comments with a response and crosswalk to where in the manuscript edits were made.
Issue |
Response |
Page/line |
2.1 Introduction: - there is no systematic review of the literature, i.e. defining individual concepts - so far conducted research on this topic, or the lack of it, which can also be shown by analyzing such databases as PRPQUEST or SCOPUS, etc. - some paragraphs do not have sources listed - the text of the introduction is actually not a scientific introduction, but rather a description of the research area. |
We believe we have addressed these concerns by reviewing the literature on which we are building our research. We have added missing citations. We have provided a scientific introduction. |
|
2.2 Materials and Methods - here everything is correct - hence my conclusion that the authors are practitioners rather than scientists
|
Thank you. The authors are a group of research partners composed of research scientists, practitioners, and community members. |
|
2.3 Results Clear and comprehensible description of the results, also very well presented graphically.
|
Thank you for this compliment. |
|
2.4 Conclusion Good summaries. You can add information about the possibility of using the research results or plans for the future continuation of the research topic.
|
Thank you for this good idea. We described how our future research will build on this current research. |
Page 16 Line 583 |

Round 2
Reviewer 1 Report
After assessing the response of the authors to the comments, it is considered that the weakest aspects of the manuscript have improved.
Author Response
October 11, 2021
International Journal of Environmental Research and Public Health
IJERPH Editorial Office
MDPI, St. Alban-Anlage 66, 4052 Basel, Switzerland
Dear Editorial Board,
I am writing in response to the reviewer’s suggestion regarding our manuscript Community Resident Perceptions of, and Experiences with, Precarious Work at the Neighborhood Level: The Greater Lawndale Healthy Work Project to the special issue Worker Safety, Health, and Well-Being in the USA.
Upon the review of the request for modifications, the reviewers suggested that could add brief language where applicable (i.e., abstract, introduction, and/or discussion/conclusion) on the link between the important paper crux and the future of work. We have done so on page 3 lines 104-110 and 120-128.
Thank you for your review of our work.
Jeni Hebert-Beirne, PhD, MPH
Associate Professor
Associate Dean for Community Engagement
Collaboratory for Health Justice Director
SPH Community Health Sciences
1603 West Taylor SPHPI 645
Chicago, IL 60612
(312) 355-0887 [email protected]
Reviewer 2 Report
All indicated areas for changes have been corrected.
Thank you
Author Response
Thank you for the feedback on our modifications based on your first round of feedback. Please also note it was suggested that we could add brief language where applicable (i.e., abstract, introduction, and/or discussion/conclusion) on the link between the important paper crux and the future of work. We have done so on page 3 lines 104-110 and 120-128.